# Attitudes to Three Weight Maintenance Strategies: A Qualitative Study

**DOI:** 10.3390/nu14214441

**Published:** 2022-10-22

**Authors:** Frances Bird, Aidan Searle, Peter J. Rogers, Clare England

**Affiliations:** 1Nutrition and Behaviour Unit, School of Psychological Science, University of Bristol, Bristol BS8 1TU, UK; 2National Institute for Health and Care Research Bristol Biomedical Research Centre, University Hospitals Bristol and Weston NHS Foundation Trust and University of Bristol, Bristol BS2 8AE, UK; 3Centre for Exercise Nutrition and Health Sciences, School for Policy Studies, University of Bristol, Bristol BS8 1TZ, UK

**Keywords:** daily weighing, diet habits, body weight maintenance, qualitative

## Abstract

Weight loss maintenance can be difficult and ultimately unsuccessful, due to psychological, behavioural, social, and physiological influences. The present study investigated three strategies with the potential to improve weight maintenance success: daily weighing, missing an occasional meal, habitually changing high energy foods. The principal aim was to gain an understanding of attitudes to these strategies in participants who had recent experience of weight loss attempts, with or without maintenance. This was a qualitative study involving semi-structured interviews, with 20 participants aged 18–67 (twelve females), analysed using thematic analysis. Most participants disliked daily weighing and missing an occasional meal for long-term maintenance and were concerned about potential negative effects on mental health. All participants had experience of habitual changes to high energy foods and regarded this strategy as obvious and straightforward. Replacement of high energy foods was favoured over elimination. Participants preferred strategies that felt flexible, “normal” and intuitive and disliked those that were thought to have a negative impact on mental health. Further investigation is needed on whether concerns regarding mental health are well founded and, if not, how the strategies can be made more acceptable and useful.

## 1. Introduction

The majority of adults living in England are obese (28%) or overweight (36%) [1]. This is despite an increase in the percentage of adults attempting weight loss and maintenance of loss [2]. The transition from weight loss to weight maintenance can be challenging. Around 60% of people may be unsuccessful at maintaining weight one year after weight loss [3]. In part, weight maintenance after weight loss is difficult because, compared with the previously higher body weight, energy expenditure is decreased and appetite increased [4,5]. Thus, weight will be regained if the individual returns to their previous dietary and physical activity habits. To maintain their lower weight, the individual must eat less despite having somewhat increased appetite and/or be more physically active, than when weight was higher [5]. However, many weight loss strategies only change behaviour in the short-term [6]. Techniques involving strict restraint—such as a very low-calorie diets and avoidance of conventional food—can be unsustainable for weight maintenance [7]. There is therefore a need for interventions that target weight loss maintenance. A first step is to identify strategies that could be incorporated into such interventions and determine whether they are perceived to be practical, acceptable, and potentially useful by the target group.

Recent research has identified three behavioural strategies (Table 1) that could be incorporated into successful weight maintenance programmes [8]. 

One of these strategies, reducing intake of energy-rich foods, is both widely accepted and supported by basic research [29]. By contrast, the strategies of frequent or daily self-weighing and missing meals, are more controversial. Whilst, as outlined in Table 1, they follow from evidence that healthy humans with access to more than adequate food supplies have the capacity to consume food energy substantially in excess of energy requirements [5,16], anecdotally at least, there is concern that self-weighing and/or missing meals might increase disordered eating attitudes and behaviours [30]. However, although self-weighing is more frequent among people diagnosed with an eating disorder [30], adverse psychological effects have not been shown to increase in participants randomised to daily self-weighing intervention, rather their body dissatisfaction was decreased compared to waiting-list control participants [31]. Similarly, skipping breakfast is reliably correlated with having a higher BMI [32], but skipping breakfast reduces daily energy intake and tends to reduce body weight [23]. The principal aim in this study was to gain an understanding of attitudes to these potentially useful, but also potentially controversial, strategies in relation to weight loss maintenance, in participants who had recent experience of weight loss or weight loss maintenance. This included a focus on perceived practicality, acceptability, and usefulness of the three strategies.

## 2. Materials and Methods

### 2.1. Study Design, Participants, and Recruitment

This was a pragmatic qualitative study, using individual semi-structured interviews. The study is reported using SPQR reporting guidelines (Appendix A) [33]. Eligible participants were adults (≥18 years) with experience of intentional weight loss or weight loss maintenance in the past year. People who self-reported a history of a diagnosed eating disorder or with a body mass index < 18.5 kg/m^2^ were excluded.

Participants were recruited via advertisements on a university website, a newsletter sent to people interested in opportunities to participate in research (1200 subscribers) and on Facebook to psychology students. Participants were offered an incentive of £15. Prior to interview, participants were given the opportunity to ask questions about the study, confidentiality was assured with regard to research records and data protection and it was explained that they could withdraw from the interview at any point. Written informed consent was obtained from all participants prior to interview. Post interview, participants were debriefed with further details of the study and given the opportunity to ask any further questions.

Ideally, sampling would have continued until data saturation was reached, where data saturation describes the point at which integration of new participants does not produce new findings. However, due to constraints of time and budget, the sample size was pre-determined using a calculation involving the expected population theme prevalence of the least prevalent theme, the power and number of desired instances of the theme [34]. Based on the assumption that dieting prevalence was 47.4% in 2013 in England [2] and 80% power, a sample size of 20 gives us somewhere between 5 and 10 desired theme instances. The final sample of 20 participants included university staff, students, and members of the general public.

### 2.2. Procedure

A topic guide was developed by the authors FB and AS following an assessment of literature and theory, and discussion with experts in the field of nutrition (Appendix A). Two pilot interviews (not included in analysis) were conducted and used to further refine questions and prompts. The interview began with open questions about participants’ previous weight loss or maintenance experiences and strategies they use for weight maintenance before moving on to their attitudes towards the three selected strategies.

Each strategy (Table 1) was introduced, and participants were asked to give their initial thoughts about the strategy and prompted if necessary to consider acceptability, practicality, and usefulness. Then, a brief rationale was given and, in the light of this, participants were asked to discuss the strategy further. To help explain energy-density for strategy 3 (habitually changing high energy foods) participants were shown pictures of isocaloric portions of a high-calorie food and a lower-calorie food. All interviews were conducted by FB based on the topic guide, adopting a flexible approach. A flexible approach refers to the order in which questions were asked as participants may pre-empt questions or answer questions in a way that prompts the order in which the interviewer asks subsequent questions. Exact questions, prompts and probes asked were adjusted depending on the content and depth of various discussion points with each participant. Participants were encouraged to talk freely and openly about their experiences. Interviews were conducted face-to-face in a quiet meeting room on University of Bristol premises and were between 25 and 65 min in duration and were conducted in March and April 2019.

### 2.3. Data Analysis

Audio-recordings of each interview were transcribed verbatim. NVivo Pro (Version 12.3) was used to manage and code data extracts.

Data were analysed using inductive thematic analysis [35]. Transcripts were read and re-read for familiarisation by FB and AS who independently coded a selection of transcripts and developed a final coding frame through discussion. This was applied to the remaining data set by FB. Data collection and coding were conducted concurrently in an iterative approach that continued until no new codes were identified from the data.

Next, FB and AS derived initial themes and sub-themes. Final themes were derived after further interpretation, familiarisation with the data and consideration of the aims by FB, AS and CE.

## 3. Results

Twenty participants (8 males, 12 females) aged 18–67 years were recruited. Five participants reported that they were successfully maintaining weight, ten successfully losing weight and five that they had either failed to lose weight or had regained after weight loss. Table 2 outlines characteristics of the participants, their experiences and strategies currently in use.

Four key themes were derived from the interviews: Lifestyle, Attitudes to Food, Self-Awareness and Making Long-term Changes.

### 3.1. Lifestyle

Most participants described the difficulty of balancing lifestyle alongside weight management, and this affected their attitudes to the proposed strategies. Some participants perceived daily weighing to be unrealistic and too effortful: 

“Don’t know whether I’d have the time every morning to do it”(Maddie)

For a few participants, not having scales had been a barrier to weight loss:

“Not having any scales, even if you do lose weight, you don’t notice […] if I had been able to see the difference, I would have had much more willpower to keep on doing it I think.”(Lucas)

Several participants thought missing a meal could be timesaving:

“It would make my mornings less stressful because I’ll get up at 5:50 every morning, get ready, get the dog out, sit down and try and get something to eat before going out to work. Probably make my mornings easier.”(Chloe)

Although many participants noted missing meals was often circumstantial rather than pre-planned:

“I mean, some days I have missed breakfast because I’ve been late in or, you know, there’s been traffic and I haven’t had time. Occasionally, yes, I have missed it. So, I, kind of, just carry on, and say, “Yes, well, I could do that,”(Victor)

Most participants described the added difficulty of foods being readily available and were already using a strategy of avoiding buying high-calorie foods although, for one participant, this was not successful:

“At home, I never buy cookies or cake because I know that if it’s in my cabinet or fridge, I’m going to eat it all.”(Alexa)

“When it’s not there you just think, ‘Sod it, I’ll go to the corner shop and buy something’”(Victor)

### 3.2. Attitudes to Food

All participants discussed their attitudes to food, how they impacted on weight management and would affect the proposed strategies. Two sub-themes were derived: Food as Consolation and Reward and Perceptions of Hunger.

#### 3.2.1. Food as Consolation and Reward

Participants often spoke about using food as a comfort from stressful emotions and were concerned that daily weighing might cause stress:

“you’d give yourself stress about it [daily weighing], wouldn’t it? So, what I’m trying to do is minimise the stress, which then is a trigger for eating.”(Connie)

Or that daily weighing when maintaining weight would induce guilt over food choices:

“I would feel really terrible if I wanted to eat a cake and I’m, like, “Oh, yesterday, I put weight on.” And I think cake is nice, so I don’t want to feel guilty all the time. I wouldn’t weigh myself more than once a week after losing weight.”(Nadia)

Many participants found eating high-calorie foods to be pleasurable and rewarding, which led to overeating. Consequently, experiences of and opinions on cutting out high-calorie habits and replacing with low-calorie alternatives were varied. Many participants said that replacement reduces feelings of restriction, guilt, and temptation:

“If you’re not having it, you’re not going to be tempted to say, “I’ve just had that one biscuit. I really fancy another one,” because you haven’t had it but you’ve replaced it with something else that’s sweet so that might satisfy your sweet craving.”(Julie)

However, a few participants disliked this idea because it was not as rewarding:

“I have a bad mindset, where I’ll think, “Well, if you’re going to indulge in something, you may as well have something that you really want,” that’s not a low-calorie version of it.”(Maddie)

Thus, some participants stated that cutting out specific highly desirable high-calorie habits (chocolate, biscuits, alcohol, and crisps were frequently mentioned) was preferable:

“I think if I was maintaining weight, I would still cut out chocolate, except for an absolute rare treat. I’m addicted to it and I’ll just eat as much as I can.”(Esme)

#### 3.2.2. Perceptions of Hunger

Most participants discussed hunger in relation to weight control. A perceived disadvantage of daily weighing was having to adjust behaviour in response to the weight on the scales rather than internal hunger cues:

“I’m not going to change my daily habits massively I don’t think. If I am really, really hungry, I don’t think I’m going to go to bed starving.”(Lucas)

For many participants, the idea of planning to miss a meal was disliked due to difficulty coping with hunger and a concern that hunger would drive unhealthy choices:

“I definitely wouldn’t miss a meal to try and lose weight because also the other thing I know with myself is that if I did that, like, later in the day or evening, I just eat whatever because I’d be hungry. So, it helps me to make more sensible choices if I do have something to eat three times a day and my snacks as well.”(Esme)

Consequently, to avoid making poorer choices, some participants preferred eating more frequently, rather than less:

“I can control the amount of food I eat if I eat five times a day. I can more or less control it. So, between breakfast and lunch, I need maybe an apple or something.”(Nadia)

However, other participants perceived that missing a meal could be useful in response to bodily cues, or previous overeating:

“If you know, “Right, I’m not hungry,” so if you’re not having dinner because you’re not hungry, it means your body doesn’t need it anyway.”(Julie)

“I think it would be worth bearing in mind specifically in the examples where you’ve maybe indulged in a meal more than you would normally. I think then it could be a neat little damage limitation thing.”(Paul)

The explanation that people don’t completely compensate for missed meals re-enforced some participants’ own experiences, but other people were unconvinced:

“If there are days when you’ve been busy at work and you’ve missed lunch, you don’t then go and have twice as much, do you, when you go home for dinner? So, that is true, really.”(Flora)

“I also think I would compensate. I know you said the study says you don’t, but I think I would.”(Seth)

Additionally, some individuals were conflicted about the importance of breakfast, and several were already skipping breakfast regularly:

“I’ve got this thing, “You must eat breakfast,” you know, but I’m not sure if that’s what I think.”(Chloe)

Participants with experience of replacing high-calorie foods with low-calorie alternatives liked this strategy because it prevented feelings of hunger:

“I think it’s really good. That’s the way I’ve done it. It’s really worked for me. I think it just satisfies the need to eat something”(Henry)

However, others felt it was tempting to increase portions of the low-calorie foods and simply cutting out high-calorie foods was preferable:

“I think, for me, there’s still the danger that I’d think, “Oh, it’s lower calorie, so I can have more of it,””(Sophie)

“I don’t even like the idea of eating those rice cakes, and things like that. I don’t buy their posing as the healthy alternative. I think you just tend to eat a lot of those things then”(Alexa)

### 3.3. Self-Awareness

Participants spoke about their own perceptions, beliefs and actions related to weight management. Most participants discussed self-image and self-worth in some way. Daily weighing was disliked because of adverse effects on self-worth, but still seen as potentially useful especially when combined with monitoring:

“Immediately I’m like, “Oh, that’s so critical every single day.” But then just reflecting, if I wanted to educate myself on maintaining my weight, I think it would be really useful. […] Maybe keeping a diary at the same time about… Maybe not like a food diary, but if you think that you ate over, or under, then you can reflect on.”(Josie)

Although understanding about how fluctuations on the scales did not always correspond with body weight did not prevent negative feelings:

“You can go up and down like a yoyo […] and you can get quite disheartened, but in actual fact, it’s not weight, it’s fluid”(Esme)

Some participants held certain beliefs and expectations about themselves being different from the type of person who would use the proposed weight maintenance strategies:

“Someone that is quite drastically overweight then it probably would be a good idea to cut it out. I think for just a normal every-day person that’s trying to maintain or something, maybe cutting it out completely isn’t necessarily the best idea because you’re more likely to want it.”(Jasmine)

“if people are at a heavy weight and they lose a lot, that’s very difficult to do, so probably they should weigh themselves daily.”(Nadia)

Participants spoke of their motivation. Those who were motivated tended to attribute their experience to internal factors:

“I can lose weight if I just put my mind to it”(Antonio)

For these participants, habitually replacing high energy foods was often a preferred strategy, as they perceived themselves to be resilient and dedicated. Being aware of the long-term nature of weight maintenance was a source of motivation in itself:

“I think that’s what keeps me motivated as well […]. You know that it’s not going to happen overnight or in a week. You need to keep going.”(Jasmine)

Renewing one’s original motivation—especially when intrinsic motivation had lessened—was useful for some participants:

“Just a reminder of how it was like to be overweight, I think that was pretty much my motivation”(Herman)

However, other participants were less intrinsically motivated and sought external support for weight loss:

“I’m better when people are telling me what to do, than setting my own goals”(Maddie)

“I’ve been quite focused recently about staying to a group, whatever. I think, for me, that sort of accountability really works.”(Flora)

Participants without clear intrinsic motivation or external support, were ambiguous about habitual change:

“I’m just like, “That would be such a boring life.” But then it would be a lifestyle change, and you would be consuming better food. So yes, I think it would be effective. And I know that I should do it, but can you tell? I’m a little bit like, “But it just tastes so good.””(Josie)

For some, having good intentions led to a positive outlook and success. These participants discussed discipline and restraint:

“I do achieve the results that I set out in my mind to achieve and that it’s just a question of application and patience and self-discipline”(Julie)

“If I put on a couple of pounds, I just try harder”(Sophie)

However, this was not the case for all participants as behaviour often did not match intentions and some individuals excused their food choices:

“You’re in the kitchen and there are strawberries or there’s chocolate, strawberries or chocolate, and you’re like, “I know I should be having you but, bugger it, I’m going to have you.” It’s making a conscious decision.”(Flora)

“I don’t see high calorie food as bad if it’s healthy.”(Josie)

### 3.4. Making Long-Term Changes

Weight maintenance was often discussed as a long-term endeavour. Participants wanted the strategies they used to be flexible and feel normal and balanced: 

“Fit your routine into your life, not your life into your routine”(Josie)

“I wanted to do something that I was going to be able to do, and enjoy doing, forever, rather than trying to crash or restrict for a set amount of time.”(Paul)

For this reason, daily weighing was often perceived unfavourably:

“It’s not how ordinary people are with their weight, and I suppose that’s what you want to be. You want it to be normalised”(Connie)

Some participants did think daily weighing may be useful when losing weight or in the early stages of maintenance, but not in the long-term:

“It does have an importance when maintaining weight loss, I think. But when maintaining a constant weight, I think it’s not so important. I think weighing every day is the key to loss, but maintaining weight, a constant weight, it’s not so important.”(Thomas)

There was ambiguity expressed by other participants who felt it could be more useful for maintenance but did not necessarily feel they would adopt the strategy themselves:

“I definitely wouldn’t do it if I wanted to lose weight. It would just become too much. I don’t know, even if I was trying to maintain it, I still don’t think I would do it, but it’s something I wouldn’t mind doing, if I had to.”(Lara)

One perceived advantage of daily weighing was that it could become habitual and support other healthy habits:

“If I knew that definitely I’m going to be weighing myself every day just to check that my weight is being maintained, then it would definitely make me think, “Right, in that case I’ve got to be mindful every day about what I eat.” It’s like a routine.”(Julie)

But there was a repeatedly voiced concern that, over time, daily weighing could become unhealthy and obsessive:

“Before you know it, you’re doing it every five minutes”(Seth)

“I think maybe that would probably not be very good for you, mentally. I think it would probably make you quite obsessed.”(Maddie)

This concern was expressed as a theoretical risk by participants who had not used daily weighing as a strategy but also by participants who were currently using it for weight loss:

“I literally weigh myself most mornings unless I’m in a real hurry and running late for work and it’s just compulsive weighing and it doesn’t tell me anything.”(Chloe)

However, one participant with previous experience of daily weighing did not think it was unhealthily obsessive behaviour.

“I did used to weigh myself every day. I don’t think it’s a bad thing to do. I don’t know if there is a fine line between being obsessive about it and actually just really trying to stick to your weight. I never felt I was obsessive, but I did used to weigh every day.”(Sophie)

Many people disliked the idea of acting on weight fluctuations:

“Want to get away from that yo-yo sort of thing”(Connie)

However, once the explanation was given that daily weighing could be used to notice and correct small increases in weight before they became large increases, several participants changed their minds and thought that theoretically it could be a useful long-term strategy:

“I think in terms of maintenance, yes, actually, I think that would be better. I think rather than acting on it the first day it goes up a little bit, if you have three days where they’re all going up, consecutively, then you can notice that kind of thing. I think that would be quite good.”(Henry)

However, despite the explanation, many participants felt that, as a long-term strategy, weighing every day would be too often, in part because of awareness that daily fluctuations in weight can be related to other bodily functions:

“I think that weight can fluctuate so much before all sorts of things, whether you’ve had loads to drink the day before, loads of water or not, whether you’re holding on to water, whether it’s the time of the month, whether you’ve been exercising.”(Flora)

In general, weighing weekly or fortnightly was preferred (and practiced).

“I think, maybe probably instead of every day, once every two weeks or every week. Because it’s bound to fluctuate”(Maddie)

Similarly, missing an occasional meal was perceived as a temporary technique more suited to weight loss by some participants:

“I see weight maintenance, like you said, as a long-term thing I don’t see skipping meals as a long-term answer”(Lara)

There was also concern that missing meals could lead to disordered eating. This concern was not unfounded since one participant discussed missing meals too often:

“I also think maybe it’s a bit of a slippery slope to an eating disorder Yes, I just think in my mind, it seems like the kind of thing that you start by just missing one occasionally, and then you start missing a couple and then it just gradually builds up.” (Henry)

“I skipped meals I guess, so I’ve stopped doing it that much […] at one point I just stopped having dinner.”(Herman)

However, other participants perceived the idea of occasionally missing a meal as being a positive strategy:

“That’s something I will definitely practice if I needed to do it because that would help you adjust and maintain your weight according to how you were feeling.”(Julie)

All participants discussed the importance of habits for long-term weight control. Cutting out high-calorie foods or replacing with low-calorie alternatives were common strategies that all participants had experience of and considered beneficial:

“I used to get that [Quorn tikka masala] quite a lot with some naan bread but naan bread would have quite a few calories in it. It doesn’t make my meal that much better to have it and it doesn’t fill me up more so I just stopped getting that.”(Lucas)

“I have changed my habit around is cereal for breakfast because I would always have a big bowl of something, Shreddies, Fruit n Fibre or whatever. Then I learnt that actually they are very, very sugary. Now I have porridge, just 40 g of porridge. That’s what my breakfast is now because that’s got less sugar in.”(Sophie)

Participants recognised the initial effort required before automaticity, and many participants found planning ahead was helpful and had strategies to do this:

“It is a bit of a mental challenge, but I stuck with it because I knew I wanted to achieve a result. After about the first two or three weeks, it then became more of a routine.”(Julie)

“I think lack of planning can make it harder, but then it’s easy to plan. We do most of our shopping on the internet, so it’s easy to make sure I’ve got loads of fruit and veg in the house”(Flora)

However, despite recognising that weight maintenance was a long-term endeavour, participants also discussed self-sabotage once reaching their goal—actively undermining weight maintenance attempts: 

“My ideal place. I’ve been there so many times. Soon as I get there, I will absolutely sabotage it”(Chloe)

## 4. Discussion

This qualitative study explored the attitudes of people who had recent experience of attempting weight loss or weight loss maintenance to the three proposed weight maintenance strategies. The aims were to determine perceived practicality, acceptability, and usefulness, and to understand individuals’ attitudes in the context of their lived experience of weight loss, weight loss maintenance and any regain. 

### 4.1. Daily Weighing

Most participants felt daily weighing was not a beneficial long-term weight maintenance strategy initially. They felt it was unrealistic and weekly weighing would be more helpful. However, once the theory behind daily weighing was explained, several participants did say they had changed their minds and viewed the strategy more favourably.

One main concern was that daily weighing might lead to an obsession with weight and could negatively impact self-image and mood. Whilst frequent weighing has been found to have no negative psychological effects [20], drawing attention to weight on a regular basis could be particularly problematic for individuals with existing body insecurities; previous research on adolescents found a significant correlation between depressed mood and body image over a five-year period [36]. 

Another perceived disadvantage was the concept of modifying behaviour based on daily fluctuations in weight. Rather, many participants favoured long-term weight goals (e.g., losing weight) which could be met irrespective of weight fluctuations (e.g., a period of weight gain could be discounted if cumulatively, weight was lost). Several people also thought that body weight changes can often be attributed to factors unrelated to energy balance (such as ‘hormones’ or ‘water retention’) and felt responding to these fluctuations in the short term would be unnecessary.

### 4.2. Missing an Occasional Meal

Participants had mixed responses to the idea of missing an occasional meal. Some participants felt that the negatives outweighed the positives and expressed concern that if missing a meal was used as a long-term strategy it could lead to disordered eating. Participants also anticipated feeling hungry, which they discussed in relation to being distracted, losing concentration, and making unhealthy food choices.

However, other participants were more favourable and drew on their own experiences of having skipped meals in the past without ill effects or feeling they had overcompensated. Arguably, for well-nourished individuals, perceived hunger is unrelated to significant ‘energy depletion’ [16]. Rather, when people reference hunger, they usually mean a state in which there is a desire to eat in the absence of fullness. Therefore, adjusting beliefs in this way (from a need to eat to a desire to eat) may aid weight management [22]. 

### 4.3. Habitual Changes to High Energy Foods

In general, participants wanted weight maintenance strategies that were flexible and normative so they could return to “normality” after weight loss. Participants felt that cutting out high-calorie foods or substituting low-calorie alternatives was an obvious, straightforward, and effective long-term strategy and was one that they were already using or had used for weight loss. Replacing high-calorie foods with low-calorie alternatives was often favoured above cutting out the high-calorie foods. Participants believed this would help satisfy ‘cravings’ and their desire to eat something.

A previous qualitative study found that successful weight maintainers adopted a long-term approach and exhibited self-control, unlike weight re-gainers [37]. Furthermore, individuals often experienced a shift in identity during the transition from weight loss to weight maintenance; feeling more accepted, liberated and comfortable [38]. It has been suggested that striving to feel normal again was more motivating and important for people than body weight itself [39]. In this attempt to regain normality, it is understandable that participants wished for flexibility. 

### 4.4. Implications for Future Research for the Three Strategies

Despite participants’ negative response to daily weighing, there is evidence that it is beneficial for weight maintenance success [21]. Future research into this strategy should consider individuals’ views to make it more appealing and examine whether concerns about becoming unhealthily obsessive are warranted. A study investigating the relationship between day-to-day fluctuations on the scale in relation to medium- and longer-term trends in body weight could be a useful first step. Daily weighing might be considered more acceptable if people were provided with trustworthy information on why their weight might have changed acutely and if any action were needed, through the provision of automatic personalised feedback, potentially through an app linked to scales. Existing literature demonstrates that daily weighing is more effective with feedback than without [21,40] and behaviour change techniques, such as goal setting and reviewing behavioural goals, are a common feature of weight management apps [41]. Combining these strategies with daily weighing could prove useful; providing individuals with the tools to feel more in control of their weight management.

Views were mixed regarding missing a meal. An appreciation that the absence of fullness rather than depletion of energy reserves is the major determinant of desire to eat may help in resisting the desire to eat until the next meal. An aid for this might be visualisation of eating as a process of meals periodically filling then emptying from the stomach (depicted by water in a saucepan), backed up by energy reserves (depicted by water in a bathtub) some 200 times larger than the energy content of a single meal [16]. Whilst desire to eat (’hunger’) is related primarily to the emptiness of the saucepan, missing an occasional meal, especially in response to previous overeating, will not deplete energy reserves sufficiently to significantly compromise bodily function. These dynamics also predict minimal subsequent compensation for a missed meal [16]. which generally what is observed [42]. 

Finally, all participants had experience of changing food habits and used this strategy regularly. However, they expressed difficulty translating their intentions into behaviour and often relied on external motivators. Specifically targeting the intention-behaviour gap during weight maintenance would be another area for research. Implementation intentions is one method of behaviour change which addresses this gap [43]. Individuals commit to a desired behaviour and create goal-directed responses to use in various situations. This aims to provide more explicit guidance on how to navigate the transition from old to new habits until the individual has habituated. Therefore, aggregating strategies with behaviour change methods could improve their success, usefulness, and acceptability.

### 4.5. Limitations

The study was advertised through a university campus and a newsletter targeting people with an interest in research and an incentive was offered for participation. Therefore, the sample is likely to not be representative of the general population. However, the use of incentives is common and felt to be innocuous unless there are factors such as dependency between the participant and researcher or the research is degrading, or high risk which did not apply to this study [44]. The sample size was theoretically derived, and we cannot assume data saturation was achieved but the sample size of 20 is also corroborated by research suggesting using between 12 and 26 participants in qualitative research using interviews [45] The use of photographs when discussing habitually removing or replacing high energy dense foods meant that this strategy was sometimes discussed in more detail, and this could have led to bias. However, all participants spoke of already using this strategy and were in favour before pictures were introduced. Finally, the study was deliberately limited to exploring the three proposed strategies in detail and did not explore other approaches that are used for weight loss such as portion control, daily activity, use of moderation when eating highly palatable foods, mindfulness in eating or monitoring of specific food groups and energy intake. We were therefore unable to compare attitudes to these other, common, approaches. We recognize that the three strategies are likely to form only a part of an intervention, alongside other, more conventional approaches such as increased physical activity. Other qualitative studies have found that successful maintainers report a range of strategies involving self-monitoring of weight and eating habits and setting upper weight limits. They generally use a flexible eating and exercise plan, which becomes stricter only if weight gain is observed [24].

### 4.6. Credibility and Reflexivity

The credibility of the research is supported by conducting an inductive thematic analysis in exploring attitudes to strategies to weight management. The researchers were interested in the openness of response to the topics to optimise and wanted to minimise the risk of any priming effects. There is also alignment between the research question and the richness of the data enhancing the plausibility and trustworthiness of the data resulting from this approach. The extent to which the research could be dependably replicated in similar conditions is high although there is a possibility of another researcher reaching different conclusions depending on sampling strategy and framing of questions. With regard to confirmability there is a clear link between the interpretation of the data, detailed descriptions and use of supporting quotes. The findings may also be transferred to another setting, context or group, however the sample is less likely to be representative of the general population and these findings may not be transferable to people of low socio-economic status or low educational attainment. However, taking the influence of context into consideration the respondents represented a wide age range and had varying degrees of success with weight management, the aims of the study were transparent, and caring and trusting relationships were nurtured with the participants fostering a high degree of authenticity in participant responses.

The interviewer (FB) was a young, female, psychology undergraduate with expert understanding of the weight maintenance strategies discussed. During data collection FB intended not to disclose any prior knowledge or opinions regarding nutrition and weight (besides what was on the interview schedule). Similarly, when analysing data and presenting the results, FB assumed a neutral standpoint, reflecting upon the range of opinions in the entire data set, and this stance was supported by AS to minimise the potential of introducing bias. However, FB, PR and AS all have a psychology background and CE is an academic dietitian with previous practice-based experience of weight management services, therefore the findings should be evaluated in this context.

## 5. Conclusions

This study found that in general, participants wanted weight maintenance strategies that were flexible, normative, and intuitive so they could return to “normality” after weight loss. Most participants felt that, despite understanding the rationale, the more controversial strategies of daily weighting and meal skipping were not acceptable. Daily weighing was thought to be unrealistic, had the potential to lead to an obsession with weight, and it was felt that weekly weighing would be more helpful. The negatives of missing an occasional meal outweighed the positives and some participants expressed concern that if missing a meal was used as a long-term strategy it could lead to disordered eating or poor mental health. If these strategies are to be used in interventions, further investigation is needed on whether the concerns are well founded and, if not, how the strategies can be made more acceptable and useful in the context of effective behaviour change.

## Figures and Tables

**Table 1 nutrients-14-04441-t001:** Selected strategies for weight maintenance.

Strategy	Further Explanation	Rationale from Research
Daily weighing, before breakfast.	It is a simple way to check on whether you have been eating too much or too little in relation to maintaining a healthy goal weight, and you can do something about it.	Easy to do and provides direct feedback. Weight gain is detected early and can be corrected at an earlier stage than less frequent or less sensitive forms of self-monitoring [9,10,11]. A useful empirical alternative to calorie counting which is often effortful and inaccurate [12,13,14,15]. It could elicit mindful behaviour regarding weight management [16]. There is evidence that frequent weighing can improve weight maintenance success [17,18,19] and increase cognitive restraint and other weight control behaviours [9]. Frequent weighing has been found to have no negative psychological effects but improved psychological wellbeing in overweight and obese adults [20]. Evidence for strong association with weight maintenance success [21].
Missing an occasional meal.	Recent studies have shown that after missing a meal you don’t end up fully compensating with what you eat later.	Missing an occasional meal could help decrease energy intake in response to potential weight gain, without constant restriction [22]. Energy intake could be maintained overall because the satiating effect of a meal is relatively short-lived. By lunchtime, a person who has skipped breakfast will have a similarly empty stomach irrespective of whether they ate breakfast 4 h previously [5,16]. Studies have examined regular meal skipping, with breakfast the best studied meal. A systematic review on breakfast skipping [23] concluded that there was no evidence that skipping breakfast promotes weight gain and cautioned against advising people that eating breakfast could be a weight loss strategy. There is little evidence for or against skipping other meals. However, irregular meal skipping, particularly when a previous meal or a previous day’s intake has been unusually large, may be a useful strategy for weight maintenance.
Habitually reducing/cutting out specific high energy foods either by substituting a high energy food for different low energy food (low fat yogurt vs. chocolate) or replacing very high calorie foods with less energy dense alternatives (feta vs. cheddar).	Another idea would be, instead of cutting it out, replacing the high-calorie food with a low-calorie alternative.	Establishing new, healthier habits is an important technique employed in successful weight management [24]. Habit-based interventions can lead to clinically significant weight loss compared to control groups, without needing explicit advice on habit formation, or an intervention from a health professional [25,26]. Replacing specific high calorie foods is likely to be straight-forward to implement and intuitive but new habits do require repeated conscious effort before becoming automatic [27]. If continued for the long term, this could be an effective and feasible proposal for weight maintenance [28].
	The idea behind cutting out or replacing a high calorie habit is that it will get easier over time, the more you do it. However, it may be difficult to start or do when in a state of hunger.	
	(Three pictures of food of energy dense vs. less energy dense food shown. Plain pasta vs. chocolate; low fat yogurt vs. chocolate; apple vs. chocolate biscuit.)	

**Table 2 nutrients-14-04441-t002:** Participant Characteristics.

Pseudonym	Gender, Age (Years)	Experience and Strategies
Antonio	Male, 61	Successfully losing weight-more walking, regular meal pattern (and not eating late), less takeaways.
Herman	Male, 20	Successfully maintaining (and trying to gain muscle)-stopped eating sweets, chocolate, weekly/twice-weekly exercise.
Henry	Male, 18	Successfully maintaining-eating healthily (home-cooked food, more vegetables, less sugar), weight training. Some-yo-yoing in weight following weight loss.
Lucas	Male, 24	Successfully losing weight (initial weight loss was somewhat unintentional but now actively trying)-calorie deficit and counting, weekly weighing.
Paul	Male, 37	Weight loss initially successful but regained and now trying to lose againweekly weighing, daily exercise, healthy eating (cutting out snacks, more vegetables). Less strict at weekends.
Seth	Male, 55	Trying to lose weight (with some success)-increased exercise, cut out red meat. Wants to improve current running ability.
Thomas	Male, 45	Successfully maintaining-was making double portions so cut down, more exercise, fairly regular weighing.
Victor	Male, 35	Tried to lose weight (unsuccessful). Tried various diets/lifestyle changes including 5:2 diet, trying to snack less, cutting out unhealthy food.
Alexa	Female, 35	Successfully losing weight (medical condition led to initial loss and then actively continued)-calorie restriction, monitoring food labels, stopped eating late, reducing portion sizes.
Chloe	Female, 55	Successfully losing weight due to dietary restriction caused by a medical condition (gluten free, dairy free, plant based, nothing fizzy). Tried losing weight in past but never able to maintain.
Connie	Female, 58	Successfully losing weight-joined Slimming World, more exercise. Some yo-yoing in weight.
Esme	Female, 67	Successfully losing weight-recently re-joined Weight Watchers. Some yo-yoing in weight.
Flora	Female, 47	Successfully losing weight (recently more yo-yoing/complacency)-member of Slimming World for 3 years, also vegan.
Jasmine	Female, 21	Successfully maintaining (and trying to gain muscle)-balanced diet, increasing protein, regular exercise, less strict at weekends.
Josie	Female, 24	Successfully losing weight-tried various diets but currently intermittent fasting with recipes inspired by keto diet, 4 days on and weekends off, irregular exercise. Some yo-yoing in weight.
Julie	Female, 55	Successfully lost weight, regained some and now trying to lose weight again-reduced portion sizes, cut out cakes, chocolate, not eating late, exercise. Some yo-yoing in weight.
Lara	Female, 19	Successfully losing weight-cut down fatty foods, more exercise, smaller portions/reducing calorie intake (also vegetarian).
Maddie	Female, 20	Unsuccessful weight loss attempts-most recently used a personal trainer (calorie deficit, regular exercise).
Nadia	Female, 38	Trying to lose weight for past 2 years (unsuccessful)-low carb diet, increasing exercise. Yo-yoing in weight.
Sophie	Female, 58	Successfully maintaining weight-following Slimming World recipes, exercise. Some difficulty maintaining if not doing exercise also.

## Data Availability

Due to the sensitive nature of the questions asked in this study, interviewees were told interview data would remain confidential and would not be shared. The lead author has full access to the data reported in the manuscript.

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
