# Peer review of "Attitudes to Three Weight Maintenance Strategies: A Qualitative Study"

_nutrients, 2022, doi:10.3390/nu14214441_

Round 1

Reviewer 1 Report

Thank you for the opportunity to review this manuscript. The manuscript is timely and produced great contribution to the literature. Please refer to the attachment for my feedback.

Author Response

Thank you very much for your kind words and suggestions for improvement. Please find our responses detailing changes below. All line numbers refer to the revised, clean, version.

  1. Title: Please add study design to the topic.

Response: The study title is now: “Attitudes to three weight maintenance strategies: a qualitative study”

  1. Introduction
  • Lines 49-50: Are there studies that support the statement? If so, please cite the sources.

Response: This statement is supported by reference 30 (Pacanowski, C.R., et al., Self-weighing behavior in individuals with eating disorders. International Journal of Eating Disorders, 2016. 49(8): p. 817-821), and this has been added. Line 53-54.

  • The problem statement and significance of the problem are clearly presented with supporting evidence/literature. However, the aim of the study stated in lines 56-60 is not consistent with the aims presented in lines 408-410. Were the focus “perceived practicality, acceptability and usefulness” (lines 59-60) or “perceived acceptability and feasibility” (lines 408-409)?

Response: We apologise for the discrepancy – the focus was indeed “perceived practicality, acceptability and usefulness” and this has been corrected. Line 387-388

  1. Materials and methods
  • Line 63: Was the design “pragmatic qualitative study” or phenomenology? The “Materials and Methods” section and line 409 seem to suggest phenomenology was the research method. Please clarify.

Response: The study was a pragmatic qualitative study. Phenomenology is a form of qualitative research that focuses on the study of an individual’s lived experiences within the world, however this should not be confused with the researchers’ explanation of the strategies to enhance their understanding of the approach in the context of this research.

  • Line 69: Please be specific about what social media platforms were used for recruitment.

Response: The platform was Facebook, in addition to the advertisements placed. This has been added. Line 74.

  • Lines 70-72: In qualitative studies, data saturation is typically used as a guiding principle to determine the sample size. How was this guiding principle be used in determining the sample size for the study?

Response: Due to constraints of time and budget (sadly not uncommon in research) the sample size was pre-determined. We have added the following to the methods.

“Ideally, sampling would have continued until data saturation was reached, where data saturation describes the point at which integration of new participants doesn’t produce new findings. However, due to constraints of time and budget, the sample size was pre-determined using a calculation involving the expected population theme prevalence of the least prevalent theme, the power and number of desired instances of the theme [33]. Based on the assumption that dieting prevalence was 47.4% in 2013 in England [2] and 80% power, a sample size of 20 gives us somewhere between 5 and 10 desired theme instances.” Lines 81-88

And to the limitations:

“The sample size was theoretically derived, and we cannot assume data saturation was achieved but the sample size of 20 is also corroborated by research suggesting using between 12 and 26 participants in qualitative research using interviews [44]” Lines 476-479.

  • Line 74: Please include a table to present the topic guide used for the semi-structured

Response: The full topic guide has now been included in supplementary materials.

  • Lines 80-85: The participants were asked to give their thoughts on acceptability. How was information on their perceived practicality and usefulness (also see line 59-60) gathered?

Response: Participants were prompted to consider acceptability, practicality, and usefulness if they did not spontaneously discuss these topics. We have added:

“Each strategy was introduced, and participants were asked to give their initial thoughts about the strategy, and prompted if necessary to consider acceptability, practicality, and usefulness.” Lines 99-100.

  • Line 85: “….. , adopting a flexible approach.” Please clarify what the flexible approach was referring to.

Response: This has been clarified in the text to read:

“A flexible approach refers to the order in which questions were asked as participants may pre-empt questions or answer questions in a way that prompts the order in which the interviewer asks subsequent questions. Exact questions, prompts and probes asked were adjusted depending on the content and depth of various discussion points with each participant. Participants were encouraged to talk freely and openly about their experiences.” Lines 104-109

  • Where was the interviews conducted? Did all researchers or FB conduct the interviews?

Response: FB conducted all the interviews, face-to-face, in University of Bristol meeting rooms. This has been clarified in the text.

“All interviews were conducted by FB”. Lines 103-104.

“Interviews were conducted face-to-face in a quiet meeting room on University of Bristol premises” Line 110.

  • Please elaborate on how trustworthiness (credibility, transferability, dependability, confirmability, and authenticity) in the data and analyses was established.

Response: We apologise for the omission and have added the following to section 4.6 (formerly Reflexivity):

4.6 Credibility and reflexivity:

“The credibility of the research is supported by conducting an inductive thematic analysis in exploring attitudes to strategies to weight management. The researchers were interested in the openness of response to the topics to optimise and wanted to minimise the risk of any priming effects. There is also alignment between the research question and the richness of the data enhancing the plausibility and trustworthiness of the data resulting from this approach. The extent to which the research could be dependently replicated in similar conditions is high although there is a possibility of another researcher reaching different conclusions depending on sampling strategy and framing of questions. With regard to confirmability there is a clear link between the interpretation of the data, detailed descriptions and use of supporting quotes. The findings may also be transferred to another setting, context or group, however the sample is less likely to be representative of the general population and these findings may not be transferable to people of low socio-economic status or low educational attainment. However, taking the influence of context into consideration the respondents  represented a wide age range and had varying degrees of success with weight management, the aims of the study were transparent, and caring and trusting relationships were nurtured with the participants fostering a high degree of authenticity in participant responses.”  Lines 494-511.

  • Ethical issues pertaining to human subjects
  • Please describe the consent process.

Response: We apologise for the omission and have added the consent process:

“Prior to interview, participants were given the opportunity to ask questions about the study, confidentiality was assured with regard to research records and data protection and it was explained that they could withdraw from the interview at any point. Written informed consent was obtained from all participants prior to interview. Post interview, participants were debriefed with further details of the study and given the opportunity to ask any further questions.” Lines 74-80.

  • Please provide information about IRB approval

Response: IRB approval is not applicable as this is a UK based study that was granted approval from the authors’ institution, as detailed in the Institutional Review Board Statement: Ethics approval was obtained from the School of Psychological Sciences’ Research Ethics Committee, University of Bristol (ref 79902).

  1. Results
  • It would be more effective if information from Table 2 was presented as a narrative summary in the manuscript, as opposed to including table 2.

Response: Thank you for this suggestion. We have considered it, but we believe that presenting this information as a participant characteristics table aids clarity.

  • The analytic findings were adequately supported by the findings (as presented in quotes). However, I would suggest eliminating some of the quotes that do not directly substantiate the analytic findings.

Response: Thank you for this comment. We agree that some quotes do not directly substantiate the analytic findings or provide illuminating background information. The following changes have been made:

  • We have removed quotes from the beginning of Section 3.1 which now reads:

‘Most participants described the difficulty of balancing lifestyle alongside weight management, and this affected their attitudes to the proposed strategies. Some participants perceived daily weighing to be unrealistic and too effortful. Line 134-136

  • Removed quotes relating to cost of healthier option at the end of Section 3.1

  • Section 3.2.1 have removed quotes at the beginning, which now reads:

‘Participants often spoke about using food as a comfort from stressful emotions and were concerned that daily weighing might cause stress. Line 163-164

  • Removed quotes relating to specific overeating incidences.

‘Many participants found eating high-calorie foods to be pleasurable and rewarding, which led to overeating. Consequently, experiences of and opinions on cutting out high-calorie habits and replacing with low-calorie alternatives were varied. Many participants said that replacement reduces feelings of restriction, guilt, and temptation’ Line 172-175.

  • Section 3.2.2 have removed quotes at the beginning, which now reads:

‘Most participants discussed hunger in relation to weight control. A perceived disadvantage of daily weighing was having to adjust behaviour in response to the weight on the scales rather than internal hunger cues’ Line 188-190

  • Section 3.3, have removed quotes from the beginning, which now reads:

‘Participants spoke about their own perceptions, beliefs and actions related to weight management. Most participants discussed self-image and self-worth in some way. Daily weighing was disliked because of adverse effects on self-worth, but still seen as potentially useful especially when combined with monitoring’ Line 233-236.

  • Removed quotes related to individual differences. Line 244.

  1. I strongly recommend using a standard guideline to report your qualitative research. An example is the Standards for Reporting Qualitative Research (SRQR).

Response: The SRQR has been completed and can be found in Supplementary materials.

Reviewer 2 Report

Dear Authors,

Thank you for the opportunity to review the article entitled „Attitudes to three weight maintenance strategies” which addresses attitudes to weight loss maintenance, in participants who had recent experience of  weight loss or weight loss maintenance. The introduction of the article is well developed, the authors presenting results from studies which addressed behavioural strategies used in the weight loss maintenance period, which were synthesized in three approaches: daily weighing, missing an occasional meal, habitually reducing specific high energy foods.

The subject under study is certainly important, especially because 60% of people will regain the weight in the first year after the period of weight loss. The authors should make clearer what is the gap in the literature that is filled with this study, which is a very important aspect because there are few studies on the motivations and behaviours during the period of weight loss maintenance. The article presents interesting results, however the method of establishing the sample size necessary to obtain saturation must be described in more detail in the methodology section.

The study is correctly designed and technically sound. The methods used in this research are well described and provide sufficient details to be understand. The research methodology is in line with the proposed objectives. The results are appropriately interpreted and respond to the hypothesis of the study.

The disscutions are relevant the paper's focus area and address the findings of the research in relation with other studies and also propose potential explanation on  behaviors and attitudes adopted during the period after weight loss .

Sincerely yours

Alina Popa MD, PhD

Author Response

Thank you very much for your kind words and suggestions for improvement. Please find our responses detailing changes below. All line numbers refer to the revised, clean, version.

In response to specific points raised, that

  • The authors should make clearer what is the gap in the literature that is filled with this study, which is a very important aspect because there are few studies on the motivations and behaviours during the period of weight loss maintenance.

Response: We have made the gap in the literature more explicit:

“There is therefore a need for interventions that target weight loss maintenance. A first step is to identify strategies that could be incorporated into such interventions and determine whether they are perceived to be practical, acceptable, and potentially useful by the target group.” Line 40-43.

  • the method of establishing the sample size necessary to obtain saturation must be described in more detail in the methodology section.

Response: Thank you for this comment. Due to constraints of time and budget (not uncommon in qualitative research) the sample size was pre-determined. We have added the following to the methods

“Ideally, sampling would have continued until data saturation was reached, where data saturation describes the point at which integration of new participants doesn’t produce new findings. However, due to constraints of time and budget, the sample size was pre-determined using a calculation involving the expected population theme prevalence of the least prevalent theme, the power and number of desired instances of the theme [33]. Based on the assumption that dieting prevalence was 47.4% in 2013 in England [2] and 80% power, a sample size of 20 gives us somewhere between 5 and 10 desired theme instances.” Lines 81-88

And to the limitations:

“The sample size was theoretically derived, and we cannot assume data saturation was achieved but the sample size of 20 is also corroborated by research suggesting using between 12 and 26 participants in qualitative research using interviews [44]” Lines 476-479.

Reviewer 3 Report

Thank you for your interesting manuscript! However, I have some comments for the better one.

Major comments

1.      The authors mentioned 3 weight maintenance strategies in the study among many strategies using qualitative methods. These strategies concerned with diet plan only. Weight maintenance after weight loss is the long way to be sustainable. In addition, the other factors such as physical activity, dietary intake details (food group, energy intake, etc), eating patterns and eating behaviors, etc, play a key role in weight maintenance. The authors missed these important factors. Thus, it would be effect on your results and difficult to comment and discuss for it. How did you fix for it?

2.      The data collection was performed by interviews. However, you did not mention it whether telephone or face-to-face. If you did it by telephone, you cannot assess the physical appearance of the participants because you did not have the information concerning weight of the participants. On the other hand, you offered the participants as incentive of 15 pounds. So you should keep in mind for that.

Minor comments

1.     You mentioned “FB and AS” in line 74. However, you did not mention what does it mean although the clarification was included in the last part of the manuscript.

2.     Total number of participants must be revealed in the methodology.

3.     The background characteristics of the participants would be interested as the table 1, instead of description of the name of participants.

Author Response

Thank you very much for your kind words and suggestions for improvement. Please find our responses detailing changes below. All line numbers refer to the revised, clean, version.

Major comments

  1. The authors mentioned 3 weight maintenance strategies in the study among many strategies using qualitative methods. These strategies concerned with diet plan only. Weight maintenance after weight loss is the long way to be sustainable. In addition, the other factors such as physical activity, dietary intake details (food group, energy intake, etc), eating patterns and eating behaviors, etc, play a key role in weight maintenance. The authors missed these important factors. Thus, it would be effect on your results and difficult to comment and discuss for it. How did you fix for it?

Response: Thank you for this comment. You are right that weight maintenance incorporates physical activity, dietary intake, eating patterns and eating behaviour but we specifically set out to examine attitudes to these three strategies, two of which are often controversial. We recognise that, used alone, the strategies are unlikely to be enough for weight maintenance, although they allow for observation of and rapid correction of weight gain.

We have added the following to the limitations:

“We recognize that the three strategies are likely to form only a part successful weight maintenance, alongside other, more conventional approaches such as increased physical activity. A larger study with successful weight maintainers speaking from personal experience of what has worked for them would be of value. Other qualitative studies have found that successful maintainers report a range of strategies involving self-monitoring of weight and eating habits and setting upper weight limits. They generally use a flexible eating and exercise plan, which becomes stricter only if weight gain is observed [24].” Line 487-492.

  1. The data collection was performed by interviews. However, you did not mention it whether telephone or face-to-face. If you did it by telephone, you cannot assess the physical appearance of the participants because you did not have the information concerning weight of the participants. On the other hand, you offered the participants as incentive of 15 pounds. So you should keep in mind for that.

Response: We apologise for the confusion. The interviews were conducted face-to-face. This has been clarified.

“Interviews were conducted face-to-face in a quiet meeting room on University of Bristol premises and were between 25 and 65 minutes in duration and were conducted in March and April 2019.” Line

We are aware that there are potential issues in the use of incentives, but the practice is common and does not present problems in many cases. We have added the following to the limitations:

“The use of incentives is common and felt to be innocuous unless there are factors such as dependency between the participant and researcher or the research is degrading, or high risk which did not apply to this study [44].” Line 473-476.

(Ref 44: Grant, R.W. and J. Sugarman, Ethics in Human Subjects Research: Do Incentives Matter? The Journal of Medicine and Philosophy: A Forum for Bioethics and Philosophy of Medicine, 2004. 29(6): p. 717-738.)

Minor comments

  1. You mentioned “FB and AS” in line 74. However, you did not mention what does it mean although the clarification was included in the last part of the manuscript.

Response: We apologise for the confusion and have added “The authors FB and AS” Line 91

  1. Total number of participants must be revealed in the methodology.

Response: The number of participants has been added to the methods section. Line 88

  1. The background characteristics of the participants would be interested as the table 1, instead of description of the name of participants.

Response: We apologise that we are unclear as to what is being asked here. The background characteristics are listed in table 2.

Round 2

Reviewer 3 Report

Thank you for your updating manuscript!